# Computer-based quantitative image texture analysis using multi-collinearity diagnosis in chest X-ray images

**Antonio Quintero-Rincón**[1,2]☯*, **Ricardo Di-Pasquale**[1]☯,
**Karina Quintero-Rodríguez**[3]☯, **Hadj Batatia**[4]☯

**1** Department of Data Science, Data Science and AI Laboratory, Catholic University of Argentina (UCA), Buenos Aires, Argentina, **2** Department of Computer Sciences, Catholic University of Argentina (UCA), Buenos Aires, Argentina, **3** "Prof. Dr. Juan P. Garrahan" Pediatric Hospital, Medical Image Department, Buenos Aires, Argentina, **4** MACS School, Heriot-Watt University, Dubai, United Arab Emirates

☯ These authors contributed equally to this work.

* antonioquintero@uca.edu.ar

**Data availability statement:** All chest X-ray images files are available from the Kaggle public

## Abstract

Despite tremendous efforts devoted to the area, image texture analysis is still an open research field. This paper presents an algorithm and experimental results demonstrating the feasibility of developing automated tools to detect abnormal X-ray images based on tissue attenuation. Specifically, this work proposes using the variability characterised by singular values and conditional indices extracted from the singular value decomposition (SVD) as image texture features. In addition, the paper introduces a "tuning weight" parameter to consider the variability of the X-ray attenuation in tissues affected by pathologies. This weight is estimated using the coefficient of variation of the minimum covariance determinant from the bandwidth yielded by the non-parametric distribution of variance-decomposition proportions of the SVD. When multiplied by the two features (singular values and conditional indices), this single parameter acts as a tuning weight, reducing misclassification and improving the classic performance metrics, such as true positive rate, false negative rate, positive predictive values, false discovery rate, area-under-curve, accuracy rate, and total cost. The proposed method implements an ensemble bagged trees classification model to classify X-ray chest images as COVID-19, viral pneumonia, lung opacity, or normal. It was tested using a challenging, imbalanced chest X-ray public dataset. The results show an accuracy of 88% without applying the tuning weight and 99% with its application. The proposed method outperforms state-of-the-art methods, as attested by all performance metrics.

## 1 Introduction

Image texture analysis, quantification, and recognition are active research topics in biomedical imaging, computer vision, and pattern recognition. In the biomedical context, texture arises from the micro-and-macro-structural patterns of biological tissues [1]. Physicians are trained to visually interpret texture information across various imaging modalities, such as

database https://www.kaggle.com/datasets/tawsifurrahman/covid19-radiography-database

**Funding:** The author(s) received no specific funding for this work.

**Competing interests:** The authors have declared that no competing interests exist.

radiographic X-rays. The principle behind these anatomical images is based on the differences in attenuation among tissues, which are influenced by the material's atomic number, tissue density, photon energy, and material thickness. Greater tissue density leads to increased attenuation. Chest X-rays are specifically employed to assist physicians in examining the anatomy of the lungs and heart. Pixel intensities correspond to the density of matter integrated along rays, often analysed according to their texture. Tissue attenuation in X-rays has extensively been studied in medical applications. Existing methods use various techniques to improve low contrast and low dynamic ranges to help discriminate tissues and precisely identify organs, bones, tumours, and nodules. Recent studies have focused on methods to remove [2] or amplify [3] tissue components, using multiscale Shannon-Cosine Wavelet models [4] or by adjusting parametric models based on the component attenuation, contrast, and image fusion [5]. Deep convolutional neural networks (CNN), such as VGG, ResNet, DenseNet, and DeTraC, have been applied directly [6–9], or as means of representation learning combined with conventional machine learning models [10]. Other deep learning methods have been proposed, including the triplet-constrained deep hashing [11], vision-transformer [12], the dual-ended multiple attention learning models (DMAL) [13], the centralised and federated learning [14], and Wasserstein distance and discrepancy metric [15].

In high-dimensional data analysis, such as image processing or computer vision, features can be correlated, making learning difficult. Various dimensionality reduction techniques exist, including subspace, manifold-based, and shallow and deep neural network methods. Principal Component Analysis (PCA) is a dimensionality reduction technique that belongs to subspace-based methods [16]. PCA aims to project the data onto fewer dimensions while preserving its inherent statistical patterns. Technically, PCA identifies the components along which the data matrix has the maximum variance. It has applications in data exploration, noise reduction, feature extraction, and data compression.

The most straightforward method to calculate PCA is through the conventional eigenvalue decomposition of the covariance matrix. However, Singular Value Decomposition (SVD) is the standard method used to prevent computational and numerical issues. SVD is a matrix factorization technique that decomposes a complex matrix $X$ into the product of three matrices $X = U\Sigma V^T$, where $U$ and $V$ are orthogonal matrices containing the left and right singular vectors, respectively, and $\Sigma$ is a diagonal matrix with singular values indicating the importance of each component. Principal components are extracted by selecting the desired number ($k$) of singular values and the corresponding columns of $V$ ($V_k$). The lower-dimensional representation of $X$ can then be computed $X' = XV_k$. For more details about PCA and SVD, we refer the reader to [17–19].

This work proposes analysing the variability of tissue attenuation as a texture phenomenon in X-ray images using singular value decomposition (SVD). Singular values and conditional indices are proposed as textural features used in a multiclass learning model to classify clinical chest X-ray images as normal, COVID-19, viral pneumonia, or lung opacity. Note that the conditional indices of a matrix are derived using the well-known SVD method [20]. They are usually used in regression methods to diagnose multi-collinearity problems [21]. They have never been used as characteristics of texture.

SVD is commonly used in image texture analysis, where the spatial arrangement of grayscale pixels in a neighbourhood is considered to characterize phenomena present in the image. This technique is used to solve problems related to segmentation, classification, and synthesis by using statistical, structural, model-based, and transform-based methods. These methods extract textural properties to describe image texels. Texels are texture units arranged in ways that can be characterized by specific feature descriptors, which in turn use texture

operators [1]. The most frequently used texture descriptors relate to coarseness, homogeneity, density, fineness, smoothness, linearity, directionality, granularity, and frequency [22]. Table 1 summarises the most popular textural operators.

**Table 1. Texture operators.**

| Texture Operator | Statistical methods | Local binary patterns (LBP) | Fractal | Convolutional | Transform-based methods |
|---|---|---|---|---|---|
| Definition | Statistical methods that calculate properties of the spatial relationships among pixels. | A binary operator measures the texture pattern by comparing each neighbouring pixel inside a kernel neighbourhood with the central pixel. | A spatial distribution of local image textures that exhibit complex patterns in irregular shapes. | Filters whose response maps depend linearly on the input texture function. | A mathematical function that maps the intensities of the pixels of an image from the spatial domain to the transformation domain. |
| Approaches | First-order and second-order statistics: Gray-Level Concurrence Matrices (GLCM), Markov Random Field (MRF). Higher-order statistics: Gray-Level Run-Length Matrices (GLRLM), Gray-Level Size Zone Matrices (GLSZM), Surrounding Region Dependency Matrix (SRDM) | | Box-counting, Wavelet decomposition, Multiplicative cascades. | Circularly/spherically symmetric filters: Laplacian of Gaussian (LoG) filters. Directional filters: Maximum Response 8 (MR8), Gabor wavelets, Histogram of Oriented Gradients (HOG), Riesz transform. Learned filters: Steerable Wavelet Machines (SWM), Dictionary Learning (DL), Deep Convolutional Neural Networks (CNN). | Fourier Transform, Wavelet Transform, Gabor Transform, Ridgelet Transform, Curvelet Transform, Radon Transform, Discrete Cosine Transform, Scale Invariant Feature Transform (SIFT), and Ranklet Transform. |
| Operator | Nonlinear. | Nonlinear. | Depends on the type of transformation of the pixel values. | Linear: LoG, Gabor wavelets, DL, CNN. | Depends on the type of transformation of the coefficient values. |
| Advantages | Texture functions are straightforward to implement in 2D and 3D. | It is robust to illumination variations. | Fractal geometry can characterise the complexity of texture composition as tissue patterns are often repeatedly defined by other subpatterns, occurring according to some statistical rules. | They can focus on texture alone and do not include any intensity information, thus improving the robustness of the texture operator responses to variations in illumination. | The transforms are fast and provide instantaneous localization in the spatial domain. Dimensionality reduction reduces redundancy and increases the discrimination of bands by energy of interest. |
| Limitation | Poor preservation of image scales. Requires drastic reductions in grayscale levels. | The local binarisation operation results in an important reduction of the values analysed. | Fractal dimension is invariant to image scale, leading to the risk of regrouping tissue structures of different natures. | Circularly/spherically symmetric filters depend on the radial polar coordinate, thus presenting a complete lack of directional sensitivity and their invariance to local rotations. | It may require different scales and orientations, resulting in huge dimensions. They may poorly capture edges and curves in images. May to have shift-invariance limitations. |
| Features | Pixels in a neighbourhood. | Pixels in circular neighbourhoods. Each binary value is multiplied by the corresponding weight to yield a texture unit number. The feature for a texture unit is obtained by adding all the multiplications. | Degrees of self-similarity at different scales of the irregular shapes. | Depends on the map from the input texture function. | Spatial-energy relationships or transitions between pixel coefficient values. |

For experimentation, singular values and conditional indices were used to analyse the variability of tissue attenuation of X-rays in cases of COVID-19, viral pneumonia, lung opacity, and normal. The aim is to derive that the two features characterise this variability well and can be used to discriminate against COVID-19 cases.

Coronavirus disease 19 (COVID-19) is caused by a severe acute respiratory syndrome called coronavirus 2 (SARS-CoV-2) infection. Infected individuals have been reported with typical clinical symptoms involving fever, non-productive cough, myalgic, shortness of breath, and normal or decreased leukocyte count. Severe cases of infection cause pneumonia, severe acute respiratory syndrome, multi-organic failure, and death [23]. It is well-known that false-positive diagnoses often lead to more expensive follow-up tests and patient anxiety, while false-negative diagnoses may result in death if treatable conditions are not identified. It is essential to know that chest X-rays are a crucial tool for diagnosing lung infections in the medical field. The real challenge is identifying the diagnosis when an opacity is seen on the X-ray. Such an opacity could indicate bacterial pneumonia, viral pneumonia, COVID-19, or other causes of opacities (including pulmonary embolism, pulmonary oedema, pleural effusion, or lung cancer). Radiologists and pulmonologists use various imaging features to differentiate these conditions, but with generally variable results. The problem can be detected, but the precise nature of the issue would remain unclear, requiring follow-up steps such as computed tomography (CT) scans, sputum analysis, comparison with medical records, concordance with clinical symptoms, bronchoscopy, and biopsy.

It is well known that experts in biomedical imaging may have varied interpretations and potential errors. For example, symptoms of COVID-19 are very similar to viral pneumonia, potentially leading to misdiagnosis. This ambiguity is caused by the much more significant variations that occur during the texture mapping process when clinicians establish links between the visual observation of image patterns and the underlying cellular and molecular structures. These variations are partly due to the diversity of human biology, anatomy, image acquisition and reconstruction protocols, compounded by observer training. Therefore, early diagnosis using chest X-ray imaging can be crucial to avoid false diagnoses and delays in treatment, which can lead to additional costs, effort, and risks.

Typical abnormal findings of COVID-19 pneumonia reported that, in chest X-ray, parts of the lungs appear as "normal well-aerated parenchyma", which is associated with areas of accentuation of the pulmonary interstitium characterised by fine linear structures representing the foci of pneumonia, in addition to the so-called ground-glass opacities, typical of COVID-19 infection. Since these findings are among the first radiological manifestations of COVID-19 pneumonia, one can hypothesise that an accurate X-ray reading could help in the early/initial diagnosis of COVID-19 pneumonia on a routine basis, also providing the differential diagnosis with other non-COVID-19 pneumonias [24]. The progress of the disease course in some patients can be relatively rapid or plodding. Usually, the most significant involvement in COVID-19 is mainly perceived in the lower lung lobes, which evolve late into areas of pulmonary consolidation, predominantly peripheral, cloudy lung-like with bilateral pleural effusion [25]. Although COVID-19 pneumonia and other viral cases of pneumonia share similar radiographic findings, it was found typical in non-COVID-19 viral lung infections that the accentuation of the pulmonary interstitium is often at the bilateral parahilar level with a progressive extension towards the periphery.

Although chest radiography is not sensitive enough to detect ground glass opacity, which is the primary imaging feature of COVID-19 pneumonia, chest X-ray imaging is the preferred method for follow-up COVID-19 pneumonia patients admitted to intensive care units [26]. Computed tomography (CT) is another imaging method that provides better accuracy and details. However, it implies a higher radiation dose: while a standard chest X-ray

has a $0.02mSv$ effective dose (in adults), a CT chest study has a $7mSv$ effective dose [27]. Furthermore, it is not recommended that a chest CT scan be performed on frail patients in intensive care. Chest X-ray allows physicians to easily estimate the extent of alveolar opacity caused by infection, with the lowest radiation rate. When interpreting chest X-rays, it is essential to recognise particular clutter, artefacts, and ambiguities that can make diagnosing pneumonia difficult, especially when using artificial intelligence (AI) tools for automated detection. Motion artefacts, for example, can create pseudo-opacity due to the superposition of anatomical structures, particularly at the lung bases, such as the diaphragm. In addition, other artefacts are produced by the structures of the chest when X-rays are passed through. A typical case is the starburst effect around the coastal arches, generated by the high bone density of these structures. Correct identification and management of these artefacts are essential to avoid false positives and optimise the accuracy of AI systems in detecting pneumonia.

Therefore, addressing the texture analysis based on tissue attenuation in chest X-ray images is complex and challenging. This work starts from the reasonable assumption that *COVID-19, viral pneumonia, and lung opacity have tissues that exhibit a significantly different attenuation of X-rays compared to normal cases. More precisely, the hypothesis is that the relation between the singular values and the conditional indices of the image characterises this difference. These two quantities can, therefore, be used as texture features to classify COVID-19, viral pneumonia, lung opacity, and normal X-ray images.* Note that this approach fits within the category of transform-based methods, see Table 1.

The main contribution of this work is an original approach to image texture analysis using singular values and conditional indices. The underlying idea is to show that the phenomenon of near-collinearity [21] characterises the variability of attenuation in X-ray images. As an application, we design an algorithm for quantifying the variability of tissue attenuation in chest X-ray images. The algorithm estimates the proportions of the singular values and the condition indices. These are used as features to classify images as COVID-19, viral pneumonia, lung opacity, or expected. To our knowledge, this approach has not been investigated before for texture analysis in chest X-ray images. In addition, a tuning weight parameter derived from the variance-decomposition proportions is proposed to improve the multiclass classification. Precisely, the estimation of this parameter relies on the coefficient of variation derived from the minimum covariance determinant. The determinant is calculated using the bandwidth parameter $h$, which comes from the non-parametric empirical distribution of the variance-decomposition proportions. Singular values and conditional indices are multiplied by this weight to account for the attenuation variability. We will show experimentally that this tuning mechanism improves the detection of respiratory syndromes.

The remainder of the paper is organised as follows. Sect 2 presents the experimental dataset (Sect 2.1) and the proposed method (Sect 2.2), where, i) we review singular value decomposition (Sect 2.2.1), ii) introduce the Likelihood-based Scree plots (Sect 2.2.2, iii) present the non-parametric empirical distribution (Sect 2.2.3), iv) propose the coefficient of variation from the minimum covariance determinant (Sect 2.2.4), v) describe the derived features (Sect 2.2.5), vi) present the *ReliefF* feature selection algorithm (Sect 2.2.6), vii) describe the multiclass classification scheme (Sec. 2.2.7), and finally, vii) review the partial dependence analysis method (Sec.2.2.8). Experimentation using chest X-ray images is presented in Sect 3, and the results are discussed. Finally, Sect 4 draws conclusions and future works.

## 2 Material and methods

### 2.1 Database

The Kaggle public database *COVID-19 Radiography Dataset* [28] was considered for experimentation purposes. The dataset consists of chest X-ray images with 3616 COVID-19 positive cases, 6012 lung opacity cases (non-COVID lung infection), 1345 viral pneumonia cases (non-COVID infection) along with 10192 normal cases. It was compiled from Qatar University - Doha, the University of Dhaka - Bangladesh, and their collaborators from Pakistan and Malaysia. Figs 1 and 2 show image sets of COVID-19, pneumonia, lung opacity, and healthy patients. See [29,30] for more information on the complete dataset.

This dataset has many challenging characteristics. For example, in some cases, the images seem to be either viral pneumonia, bronchitis, or normal. The issue arises from the fact that the lung conditions of COVID-19 patients are particularly severe and pronounced, as shown in Figs 1 and 2. For instance, in Fig 1(a), there is a typical case showing no infiltrates or pulmonary opacities, with adequate aeration of both lung fields. Fig 1(b) shows a suspicious COVID-19, with an alveolar-interstitial pattern in the upper, middle, and lower fields (predominantly left side). In Fig 1(c), there is a viral pneumonia suggestive image (non-covid) with a diffuse bilateral parahilar interstitial pattern. Fig 1(d) shows a non-pneumonia opacity, with slight accentuation of the bilateral parahilar peribronchovascular interstitial network predominantly on the right. These are the best-case scenarios in their respective classifications, indicating the severity of the COVID-19 cases.

Similarly, Fig 2(a) shows a normal adult chest X-ray. The right perihilar reinforcement appears normal, caused by the pulmonary hilum being more exposed than the contralateral left hilum. In Fig 2(b), a patient is lying down for a chest posteroanterior (PA) view due to insufficient breathing, with numerous monitoring leads attached. In both lung fields, ground-glass interstitial infiltrates are observed at the vertices and bases that are usually linked to COVID-19. Fig 2(c) shows a standing patient for a chest posteroanterior (PA) view with relatively clean lungs, while a right parahilar interstitial infiltrate is present, representing indirect signs of air trapping. In Fig 2(d), a lack of aeration can be seen in both lung fields, where it is impossible to specify the predominant pattern/infiltrate or a particular aetiology. These represent the worst-case scenarios in their respective cases, suggesting that the normal cases are less severe. In essence, the dataset displays a clear difference in disease severity between COVID-19, viral pneumonia, lung opacity, and normal chest X-ray images.

### 2.2 Methodology

This work aims to find an original interpretable method to quantify the variability of tissue attenuation in chest X-ray images. The underlying idea is to assess the near dependencies among the image columns by calculating the large conditional indices. As mentioned, chest X-ray images of COVID-19, viral pneumonia, lung opacity, and normal cases are used for experimentation. For this purpose, a method composed of three stages is proposed (Fig 3). The first stage consists of applying the singular value decomposition to the X-ray image to obtain three parameters, namely the singular values ($\zeta$), the conditional indices ($\eta$), and the variance-decomposition proportions matrix ($\Sigma$). In the second stage, a non-parametric empirical distribution is used as a dimension reduction of the variance-decomposition proportions matrix $\Sigma$. Next, the coefficient of variation based on the minimum covariance determinant is calculated from the bandwidth vector yielding a single weight. According to our hypothesis (see Introduction 1), this parameter represents tissue attenuation that leads to a near-collinearity phenomenon. It is used to tune the weight of the features from stage one.

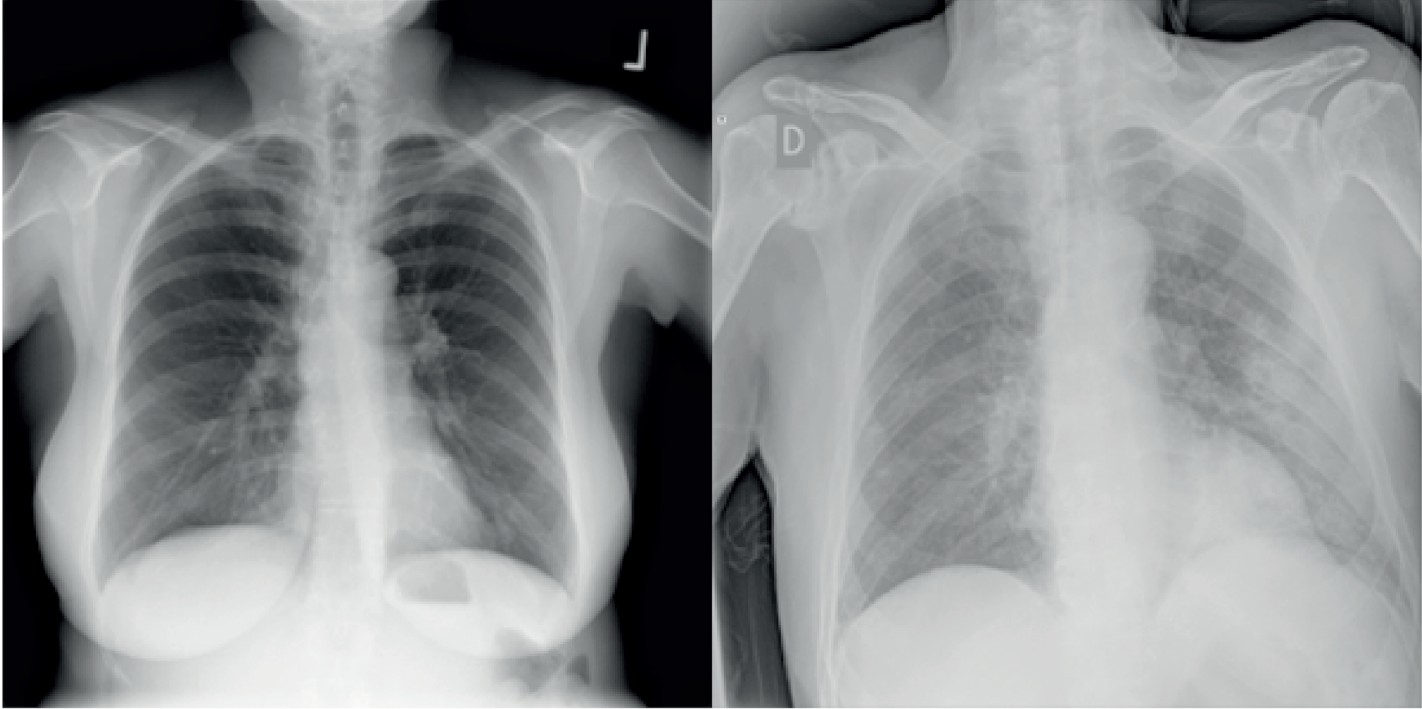

(a) Normal X-ray

(b) COVID-19

(c) Non-COVID-19 viral pneumonia

(d) Non-COVID-19 lung opacity

**Fig 1. Examples of images from best-case scenarios of the four categories in the dataset: (a) Normal chest; (b) COVID-19; (c) Non-COVID-19 viral pneumonia; (d) Lung opacity.**

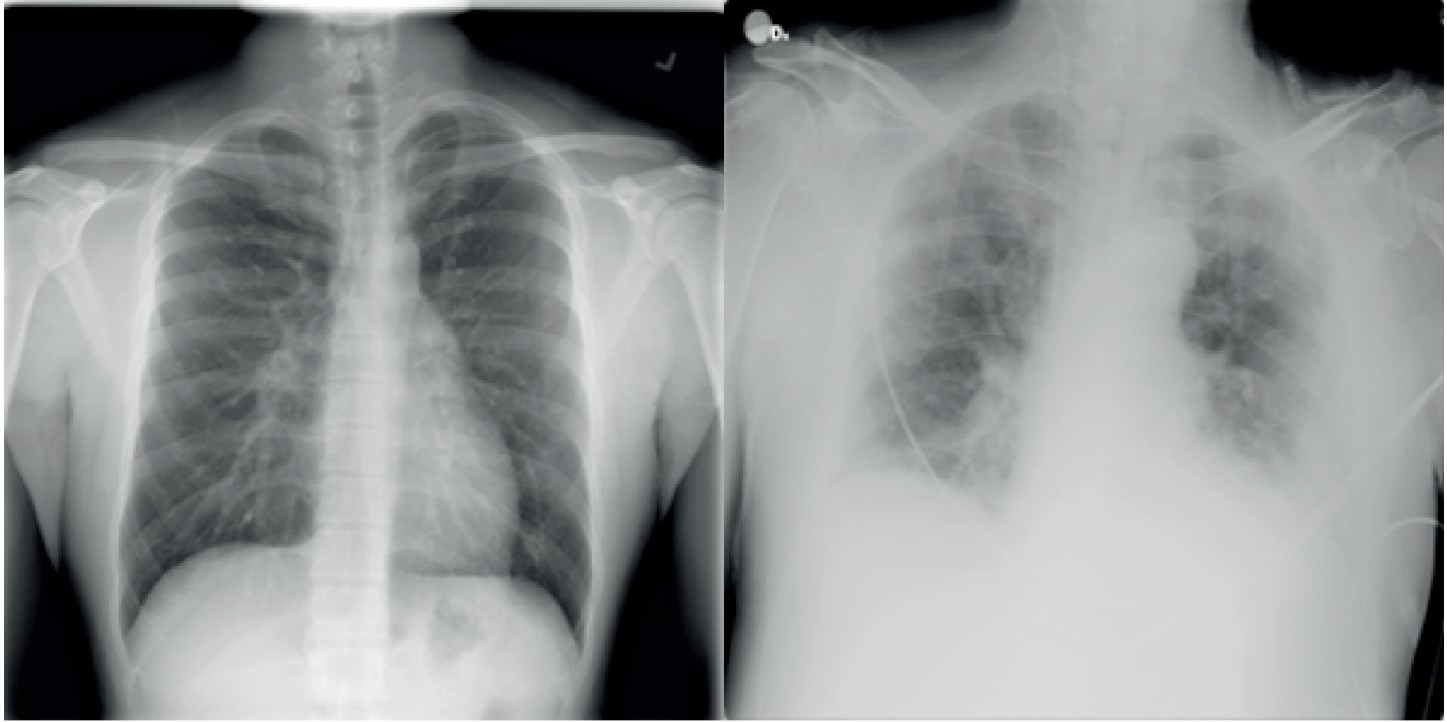

**(a)** Normal X-ray **(b)** COVID-19

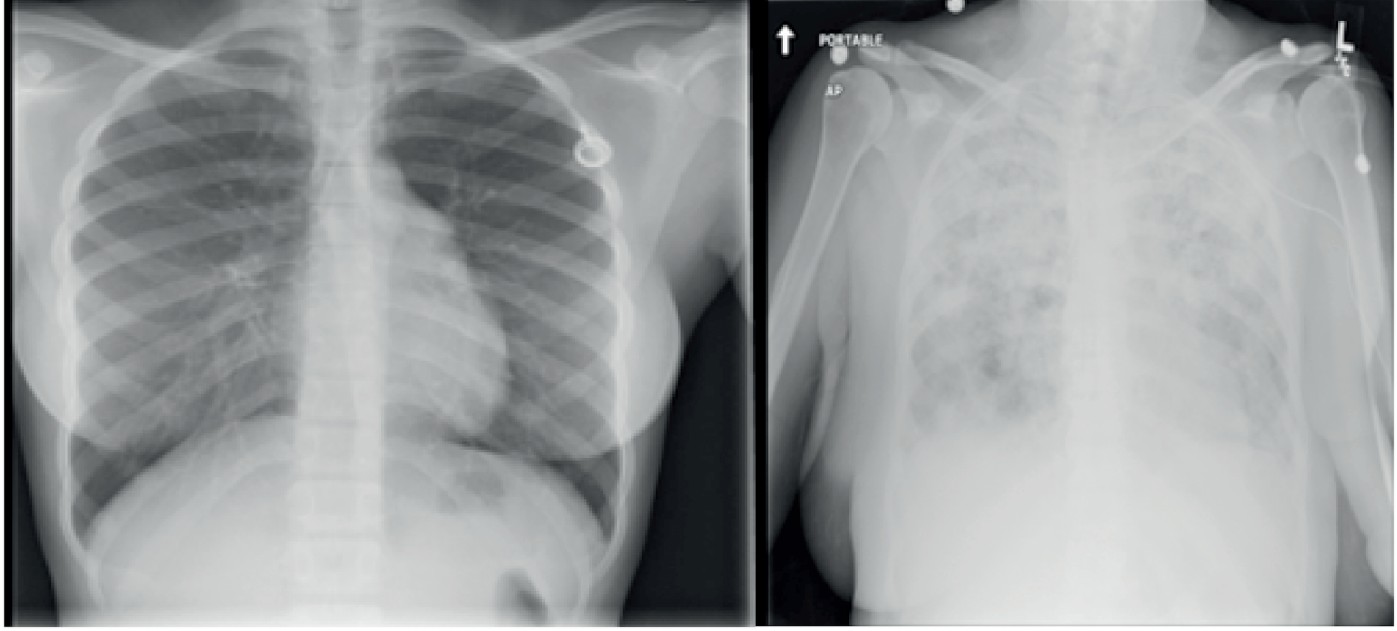

**(c)** Non-COVID-19 viral pneumonia **(d)** Non-COVID-19 lung opacity

**Fig 2. Examples of images from worst-case scenarios of the four categories in the dataset: (a) Normal chest; (b) COVID-19; (c) Non-COVID-19 viral pneumonia; (d) Lung opacity.**

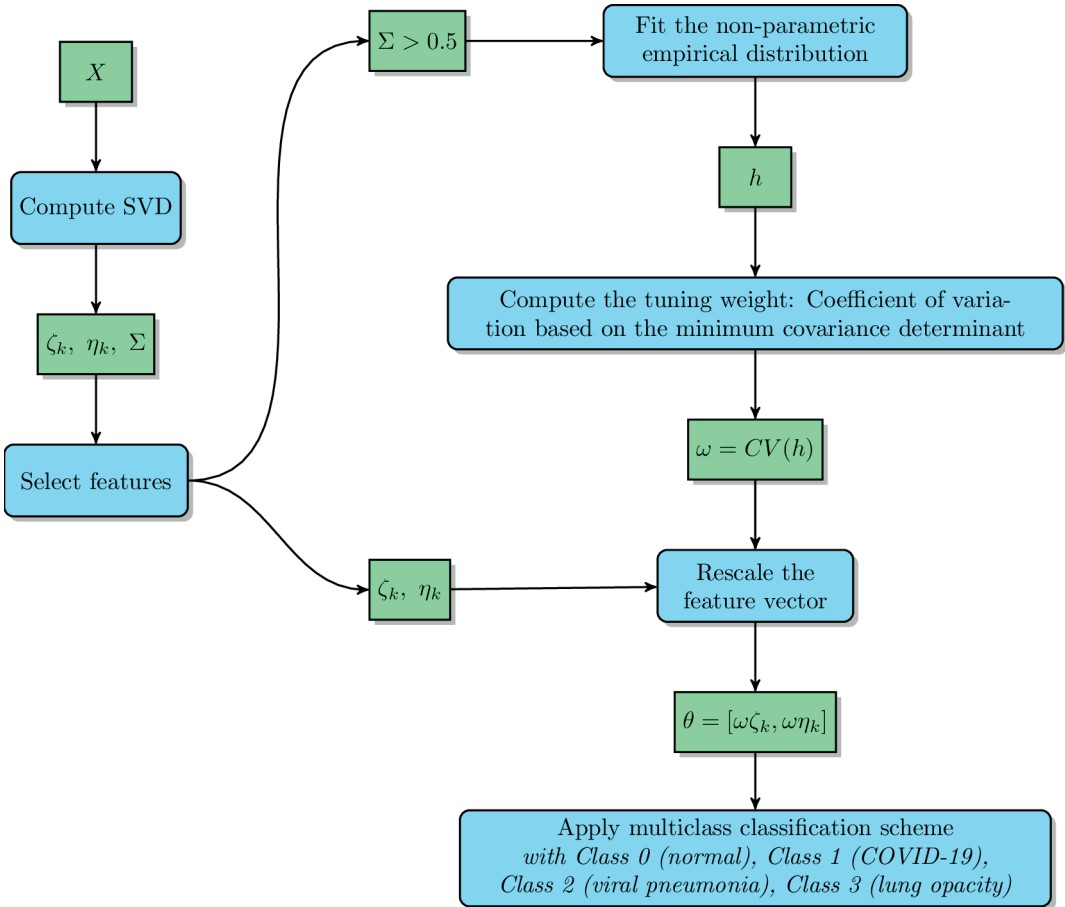

**Fig 3. Block diagram of the proposed method illustrating the processing workflow.**

More precisely, the singular values and the conditional indices are multiplied by this weight to obtain our final feature vector. In the final stage, a classification scheme is used with this feature vector to distinguish COVID-19, viral pneumonia, lung opacity, and normal cases. These three stages are detailed in the following sections.

**2.2.1 Singular value decomposition**   Let $X \in \mathbb{R}^{N \times K}$ be the known chest X-ray image matrix, where $N$ is the number of rows considered here as the number of observations, and $K$ is the number of columns viewed as the number of variables. By using the classical singular value decomposition (SVD) [31], $X$ can be expressed as $X = USV^T$, where $U \in \mathbb{R}^{N \times N}$, $V \in \mathbb{R}^{K \times K}$, $U^T U = V^T V = I_K$, and $S \in \mathbb{R}^{N \times K}$ include the non-negative diagonal elements, representing the sorted singular values of $X$.

$$\zeta_k = [S_1 \geq S_2 \geq \cdots \geq S_K] \tag{1}$$

with $S_1$ being the largest singular value, and $\zeta \in \mathbb{R}^{K \times 1}$. The conditional indices identify the number and strength of any near dependencies between variables and are given by

$$\eta_k = \frac{S_1}{S_k}, \quad k = 1, \cdots, K, \tag{2}$$

where $1 \leq \eta_k \leq \frac{S_1}{S_K}$, and $\eta \in \mathbb{R}^{K \times 1}$. Note that the larger $\eta_k$ is, the stronger the corresponding near-linear dependence.

The variance-decomposition proportions matrix, related to $V$ and $S$, is given by

$$\Sigma = \sigma^2 V S^{-2} V^T, \tag{3}$$

where $\sigma^2$ is a given variance parameter, and $\Sigma \in \mathbb{R}^{K \times K}$.

Computing the SVD decomposition is the main computational cost component in the proposed method. When applied to large dimension ($M \times N$) images, direct SVD decomposition is costly given its $\mathcal{O}(MN \min(M, N))$ complexity. In this work, we used the augmented Lanczos bidiagonalization algorithm [32], implemented in the function *svds* in MATLAB˚. This algorithm computes the first $C$ singular values with a complexity of $\mathcal{O}(MNC)$, hence reducing the complexity.

**2.2.2 Likelihood-based Scree plots** The scree plot [33] is a graphical representation of the eigenvalues against the component number, used to estimate the appropriate number of principal components to retain based on the elbow or break within its graph. Because the eigenvalues decrease monotonically from the first to the last value, a breaking point is created, though not in all cases, which suggests that the waveform begins to level off, see Fig 4. Note that this break or elbow allows choosing the number of significant components, and its dimensions can be estimated automatically using the profile likelihood function [34]. Components that appear before the break or elbow are assumed to be significant and are retained for data interpretation. In contrast, the components that appear after the break or elbow are assumed unimportant and are therefore not retained. Scree plots are helpful when there is an apparent significant deviation in the variation explained by the components [35]. For a comprehensive treatment of the scree plot, read [36,37].

The profile likelihood function is defined as follows. Let $\chi \in \mathbb{R}^p$ the $p$-dimensional vector that contains the ordered measurement of importance values or significant components,

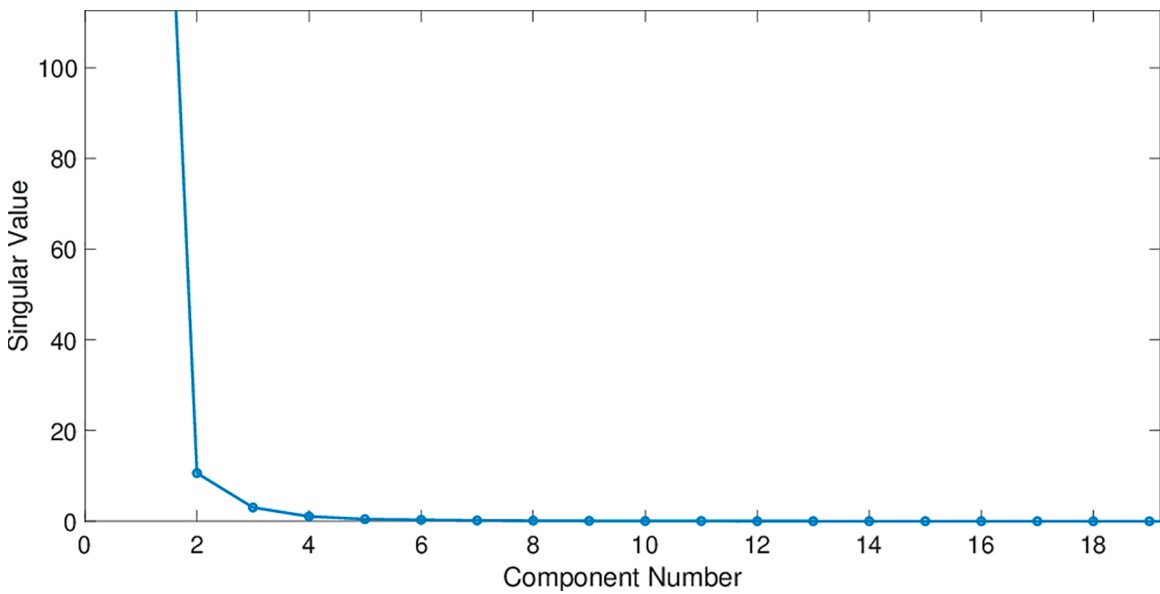

**Fig 4. Example of a Scree plot of singular values, highlighting an elbow at the second component.**

described as $d_1 \geq d_2 \geq \cdots > 0$. The problem is determining how many components to retain according to the break or elbow on the waveform. Then, for a fixed number $1 \leq q \leq p$, if a break or elbow exists at position $q$, $\mathcal{C}_1 = \{d_1, d_2, \cdots d_q\}$, and $\mathcal{C}_2 = \{d_{q+1}, d_{q+2}, \cdots d_k\}$ are defined as independent samples from two different distributions, called $f(d; \theta_1)$ and $f(d; \theta_2)$, then the log-likelihood function under the naïve independence assumption can be written as

$$\ell(q, \theta_1, \theta_2) = \sum_{i=1}^{q} \log f(d_i; \theta_1) + \sum_{j=q+1}^{p} \log f(d_j; \theta_2) \tag{4}$$

For any given $q$, the maximum likelihood estimate of $\theta_1$ and $\theta_2$ can be obtained separately from $\mathcal{C}_1$ and $\mathcal{C}_2$. By plugging these estimates into Eq (4), a profile log-likelihood for $q$ yields:

$$\ell(q) = \sum_{i=1}^{q} \log f(d_i; \hat{\theta}_1(q)) + \sum_{j=q+1}^{p} \log f(d_j; \hat{\theta}_2(q)). \tag{5}$$

Computing $\ell_q(1), \ell_q(2), \cdots, \ell_q(p)$ in Eq (5), $q$ can be estimated by maximizing the profile log-likelihood as

$$\hat{q} = \operatorname*{argmax}_{k=1,2,\cdots,p} \ell_q(k) \tag{6}$$

**2.2.3 Non-parametric empirical distribution** At this stage, we consider the distribution of the variance-decomposition proportions (i.e., the elements of $\Sigma$). This matrix is reorganised into a vector $\boldsymbol{x} = (x_1, \ldots, x_n)$. Let $\widehat{f}$ be the density estimated from observed data $x$,

$$\widehat{f}(x) = \frac{1}{n} \sum_{i=1}^{n} \mathcal{K}(x - x_i, h), \tag{7}$$

where $x_1, \ldots, x_n$ are considered random samples from the unknown distribution, $\mathcal{K}(.)$ is the kernel smoothing function, and $h$ is the smoothing parameter or bandwidth that controls the variance of the kernel.

A Gaussian kernel $\mathcal{K} \approx \mathcal{N}(0, h)$, with zero mean and standard deviation $h$, was used to build a smoothing function to represent the probability distribution of the input data.

The smoothing parameter, or bandwidth $h$, determines how the probability associated with each observation [38] (i.e., in our case, proportion) is spread over the surrounding sample space. Assuming that $f$ is a normal density function, then $h$ can be expressed as

$$h = \left(\frac{4}{3n}\right)^{\frac{1}{5}} \tilde{s}, \tag{8}$$

where $\tilde{s}$ can be estimated as follows [38],

$$\tilde{s} = \frac{mad(\log(x_i))}{0.6745} \tag{9}$$

with $mad$ denoting the median absolute deviation of the sample.

**2.2.4 Coefficient of variation based on the minimum covariance determinant** Let $X \in \mathbb{R}^{n \times p}$ the dataset matrix, where $n$ stands for the number of observations and $p$ for the

number of variables. The minimum covariance determinant (MCD) algorithm is a (uni)multi-variate location and scatter faster-robust-estimator [39]. The underlying idea of MCD is to find $h$ observations out of $n$, such that $\frac{n}{2} \leq h \leq n$, whose covariance matrix has the lowest possible determinant, therefore its mean ($\mu$) and covariance ($\Sigma$) are given by

$$\mu(MCD) = \mu(h)$$
$$\Sigma(MCD) = \Sigma(h) * C_0.$$

The parameter $C_0$, called the consistency factor, has two targets. The first is to obtain consistency of the (uni)multi-variate normal distribution, and the second is to correct for bias at small sample sizes.

The parameters $\widehat{\mu}$ and $\widehat{\Sigma}$ are estimated from the elliptically symmetric unimodal distribution of the (uni)multi-variate data as follows

$$\widehat{\mu} = \frac{\sum_{i=1} W(d_i^2) x_i}{\sum_{i=1} W(d_i^2)} \tag{10}$$

$$\widehat{\Sigma} = C_0 \frac{1}{n} \sum_{i=1} W(d_i^2)(x_i - \widehat{\mu})^* (x_i - \widehat{\mu})^T, \tag{11}$$

where $d_i = d(x, \widehat{\mu}, \widehat{\Sigma})$ is the Mahalanobis distance, $W(.) = I(. \leq \sqrt{\chi^2_{M,0.975}})$ is a weight function with $I$ as the indicator function, $*$ denotes the complex conjugate, $C_0 = \alpha/F_{\chi^2_{M+2}}(q_\alpha)$, $q_\alpha$ is the $\alpha$-quantile of the Chi distribution ($\chi^2_M$), with $\alpha = \lim_{n \to \infty} h(n)/n$, and $h$ taken such as $[(n+p+1)/2] \leq h \leq n$. To comprehensively review the MCD analysis method, consult [39–42].

The coefficient of variation (CV) ratio, based on the MCD, indicates the covariance dispersion of data from the mean value. It is given by

$$\widehat{CV} = \frac{\widehat{\Sigma}}{\widehat{\mu}}. \tag{12}$$

**2.2.5 Tuning weight** As stated above, the experimental approach taken in this work is to differentiate COVID-19, viral pneumonia, lung opacity, and normal cases in chest X-ray images by identifying the near dependencies between the image columns $X$. This identification is done by calculating the singular values and the associated large conditional indices following equations Eqs (1) and (2). Additionally, the variance-decomposition proportions ($\Sigma$)(Eq (3)) greater than a threshold of 0.5 are considered. These proportions are related to the groups of variables (i.e., image columns) involved in near dependencies. A single tuning weight $\omega$ is introduced to improve the classification for these situations. Note that $\Sigma_i$ is a vector of size $M \times 1$ with $M$ values greater than 0.5. To estimate $\omega$, the non-parametric empirical distribution is estimated (Sec. 2.2.3) for each variance-decomposition proportion $\Sigma_i$, and the vector $h = (h_1 \ldots h_n)^T$ is estimated following equations Eqs (8) and (9). The tuning weight $\omega$ is then estimated as the coefficient variation based on the minimum covariance determinant Eq (12), denoted $\widehat{CV}(.)$, of the univariate vector $h$.

$$\omega = \widehat{CV}(h). \tag{13}$$

The resulting parameter feature vector is

$$\theta = [\omega\zeta_k, \ \omega\eta_k], \tag{14}$$

where $1 \leq k \leq \Psi$. Please note that $\Psi$ are the first most representative components. Because the number of $\Psi$ components is not known in advance, the scree plot using the profile likelihood function is estimated; see Sect 2.2.2. Note that the elements of the vector $\zeta_k$ are in descending order, while the components of the vector $\eta_k$ are in ascending order.

**2.2.6 Feature relevance** The ReliefF algorithm is used to select relevant texture features for classification. This algorithm is based on a filter-method approach [43]. It works by calculating a weight $W_j$ for each feature $F_j$. Given a dataset of $n$ instances of $p$ features, the algorithm iteratively selects one random instance $x_r$. It locates each class's instance $x_q$ closest to $x_r$ and considers their feature vectors.

Initially zero, the weight $W_j^i$ at iteration $i$ is updated according to equations Eq (15) or Eq (16) depending on the class of $x_q$:

$$W_j^i = W_j^{i-1} + \frac{\Delta_j(x_r, x_q)}{m} d_{rq}, \text{ if } x_r \text{ and } x_q \in \text{ same class} \tag{15}$$

$$W_j^i = W_j^{i-1} + \frac{p_q}{1 - p_r} \frac{\Delta_j(x_r, x_q)}{m} d_{rq}, \text{ otherwise} \tag{16}$$

with

$$\Delta_j(x_r, x_q) = \frac{|x_{rj} - x_{qj}|}{\max(F_j) - \min(F_j)} \tag{17}$$

$$d_{rq} = \frac{\exp^{-\left(\frac{rank(r,q)}{\epsilon}\right)^2}}{\sum_{\ell=1}^k \tilde{d}_{r\ell}}, \tag{18}$$

where $p_r$ and $p_q$ are the prior probabilities of the classes to which $x_q$, and $x_r$ belong, respectively; $m$ is the number of iterations; $\Delta_j(x_r, x_q)$ is the difference of the value of the feature $F_j$ between observations $x_r$ and $x_q$; $x_{rj}$ and $x_{qj}$ denote the value of the feature $j$ for the observations $x_r$ and $x_{qj}$ respectively; $d_{rq}$ is a distance function such as Euclidean; $rank(r, q)$ is the position of instance $q$ among the nearest neighbours of the observation $r$, sorted by distance; $k$ is the number of nearest neighbours; $\epsilon$ is the distance scaling factor; note that for all nearest neighbours to have the same influence $\epsilon = \infty$. This feature weights procedure will allow us to assess the relative importance of the features used with our classification method. For a complete treatment of the ReliefF algorithm, read [44,45].

**2.2.7 Multiclass classification method** The final stage of our proposed method consists of using the feature vector developed above to classify X-ray images. The dataset used in this study is unbalanced. COVID-19 over normal cases has a ratio of 1:3, COVID-19 over viral pneumonia has a ratio of 3:1, and COVID-19 over lung opacity has a 1:2 ratio. The well-known ensemble learning method based on Bagged trees was considered for its good performance in various applications and for handling imbalanced data classes. The associated performance was evaluated according to the following metrics [46–48]: True Positive Rate (or recall, or sensitivity), False Negative Rate, Positive Predictive Values, False Discovery Rate, Area Under Curve, Accuracy Rate, and Total Cost. The execution time is given for comparison purposes based on a standard PC use.

**2.2.8 Partial dependence analysis** To analyse the behaviour of the classification model and assess the effect of the two features on the predictions, we performed the partial dependence analysis. This technique is used in machine learning to interpret the relationship between a single feature (or a subset of features) and the model's prediction. It helps to understand how the given feature (or subset of features) influences the predictions of the model

while averaging out the effects of the other features. It consists of creating and interpreting partial dependence plots provided in Sect 3. The technique is utilised to understand feature importance and how they contribute to predictions. It is also essential to validate the behaviour of models in healthcare applications.

Let $f(\Theta)$ denote a predictive model, with the feature vector $\Theta = (\Theta_1, \Theta_2, ..., \Theta_p)$. For binary classification, $f(\Theta)$ is the probability estimate $P(y = 1|x)$. Let $\Theta_S$ denote a small subset of $\Theta$, with its complement $\Theta_C = \Theta \setminus \Theta_S = \{\theta_{C,1}, ..., \theta_{C,N_C}\}$, of size $N_C$. The partial dependence between $\Theta_S$ and the model's output is defined as:

$$\hat{f}_S(\Theta_S) = \mathbb{E}_{\Theta_C}[f(\Theta_S, \Theta_C)], \tag{19}$$

where $\mathbb{E}_{\Theta_C}$ is the expectation taken over the marginal distribution of $X_C$ in the dataset. In practice, given the discrete nature of the data, $\hat{f}_S(\Theta_S)$ is computed as the average over all values of $\Theta_C$:

$$\bar{f}_S(\Theta_s) = \frac{1}{N_C} \sum_{i=1}^{N_C} f(\Theta_S, \theta_{C,i}). \tag{20}$$

For $K$ classes, the generalised partial dependence for class $k$ is:

$$\hat{f}_k(\Theta_S) = \mathbb{E}_{\Theta_C}[P(y = k|\Theta_S, \Theta_C]. \tag{21}$$

Eq 21 means that for each class $k$, the probability is marginalised over the complementary features to analyse how $P(y = k)$ varies with $\Theta_S$. We refer the reader to [19] for a comprehensive treatment of partial dependence analysis.

In our study, $\Theta = (\Theta_1, \Theta_2) = (\omega\zeta_k, \omega\eta_k)$ and $K = 4$. We calculated and displayed the partial dependence plots for each class to assess the effect of singular values and conditional indices on the model's decision. We performed this analysis with and without applying the tuning weight. Results and discussion are provided in Sect 3.

## 3 Results and discussion

This section reports the evaluation results of the proposed method using the previously introduced database. The dataset comprises chest X-ray images distributed as follows: $3,616$ COVID-19, $1,345$ viral pneumonia, $6,012$ lung opacity, and $10,192$ normal cases. Each chest X-ray image has a size of $N \times K$ (where $N = K = 299$). Images were converted to grayscale, retaining the luminance and eliminating the hue and saturation information.

We run two experimental scenarios. The first is without applying the tuning weights (scenario one), and the second is with the tuning weights (scenario two). For each scenario, the proposed features were used to train and test the classification model to discriminate between the four classes. For the first scenario, each image was reduced to two features $\rho_1 = [\zeta_k, \eta_k]_{k=1...\Psi}$. In the second scenario, the features were weighted using the tuning parameter $\omega$, $\rho_2 = [\omega\zeta_k, \omega\eta_k]_{k=1...\Psi}$. The scree plot method (see Sect 2.2.2) yields a boundary number of $\Psi = 5$, related to the first more representative singular values, see Fig 4. This leads to 10 features for each image, 5 times the two parameters: $[\zeta_k, \eta_k]_{k=1..5}$. Remember that $\omega$ is given by the coefficient of variation from the minimum covariance determinant of the smoothing parameters $h$ (see Sects 2.2.3, 2.2.4, and 2.2.5). For illustration, Fig 5 depicts the nonparametric distribution fitted to the variance-covariance matrix data. While the curves corresponding to the four classes appear very similar, they exhibit subtle variations that may not be

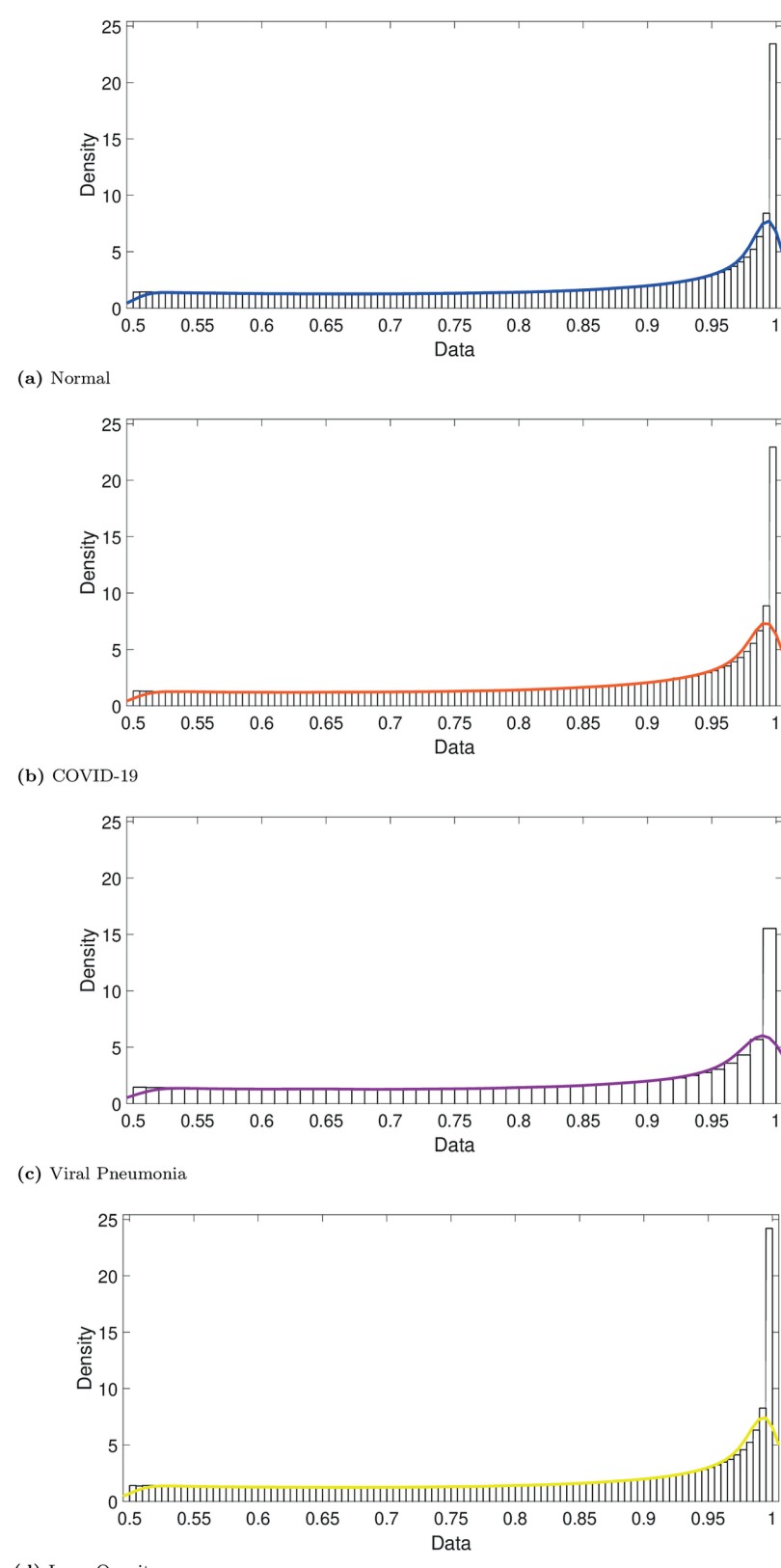

**Fig 5. Fit of the non-parametric empirical distribution to the variance-covariance matrix data: Class 0 (normal, blue dots), Class 1 (COVID-19, red dots), Class 2 (viral pneumonia, purple dots), and Class 3 (lung opacity, yellow dots).**

visually perceptible. These variations are confirmed by the statistics of the smoothing parameter $h$ (see Table 2). Although minimal, these differences are sufficient for the ensemble model to capture class-specific characteristics. The parameter $\omega$ amplifies these variations, enhancing the model's ability to learn distinctive patterns for each class.

Before applying the classification, the ability of the features to discriminate the four classes was analysed. Fig 6 shows the nonlinear scatter plots of the feature vectors over $105,825$ observations. Figs 6(a) and 6(d) show how the tuning weight parameter works. One can notice how the cases of normal (blue dots), COVID-19 (red dots), lung opacity (yellow dots), and viral pneumonia (purple dots) in the initial part, singular values ($\zeta_k$) between 0 and 9 can be easily differentiated, as shown in Fig 6(b). However, the cases are very close in the final part, as shown in Fig 6(c). The feature vector $\rho_1$ (without the tuning weight) has the next bounds. The singular values (svd) are in the interval $[0.203 \leq \zeta_k \leq 17.238]$, while the conditional indices (idx) are in the interval $[1 \leq \eta_k \leq 77.403]$. Large indices identify near dependencies among columns of $X$. Therefore, the size of the indices is a measure of how near dependencies are to detecting all X-ray images under study, especially COVID-19. The $p$-values were estimated by using the T-test method [49], with $p$-value= 0.0325 for $\zeta$, and $p$-value= 0.0001 for $\eta$, indicating high significance. Fig 7 shows the rank importance of the predictors from the ReliefF algorithm. One can notice that the conditional indices are more representative concerning the singular values without the tuning weight parameter $\omega$, as seen in Fig 6(b). All conditional indices can easily be differentiated for each case of study. However, singular values become more representative when applying the tuning weight parameter, as seen in Fig 6(d). At first, the classes are even more separated, but in the end, the separation is optimal for all cases, especially COVID-19. Note that both conditional indices and singular values were rescaled. For illustration, Table 3 shows the different thresholds for the two experimental scenarios (with and without the tuning weight parameter). The mean and standard deviation are high in the two scenarios for the COVID-19 class, which allows it to be used in a threshold detection scheme [50] to differentiate COVID-19 from the other classes under study.

For the classification stage, the dataset was randomly split into 90% ($95,243$ observations) for training and 10% ($10,582$ observations) for testing. The ensemble learning model, evaluated using 10-fold cross-validation, demonstrated promising results. Both experimental scenarios were assessed using standard performance metrics, including True Positive Rate (recall/sensitivity), False Negative Rate, Positive Predictive Value, False Discovery Rate, Area Under the Curve (AUC), Accuracy, and Total Cost. The confusion matrices, shown in the Supporting Information section in S1 Fig for the training and S2 Fig for the testing, indicate high performance across all classes with minimal false responses. S3 Fig presents the Receiver Operating Characteristic (ROC) curve and the Area Under the Curve (AUC) for both training and testing stages without the weight parameter $\omega$. All ROC curves exhibit high True Positive Rates and low False Positive Rates, with AUC values close to one for all classes. These findings suggest that the proposed model effectively distinguishes between the studied classes.

Table 4 summarises these results, highlighting that the False Negative Rate and False Discovery Rate remain low, while the tuning parameter substantially reduces the total cost.

**Table 2. Statistics of the smoothing parameter $h$ across all images and classes.**

|  | Normal | COVID-19 | Viral Pneumonia | Lung Opacity |
|---|---|---|---|---|
| **mean±std** | $0.061 \pm 0.071$ | $0.061 \pm 0.070$ | $0.061 \pm 0.044$ | $0.067 \pm 0.092$ |

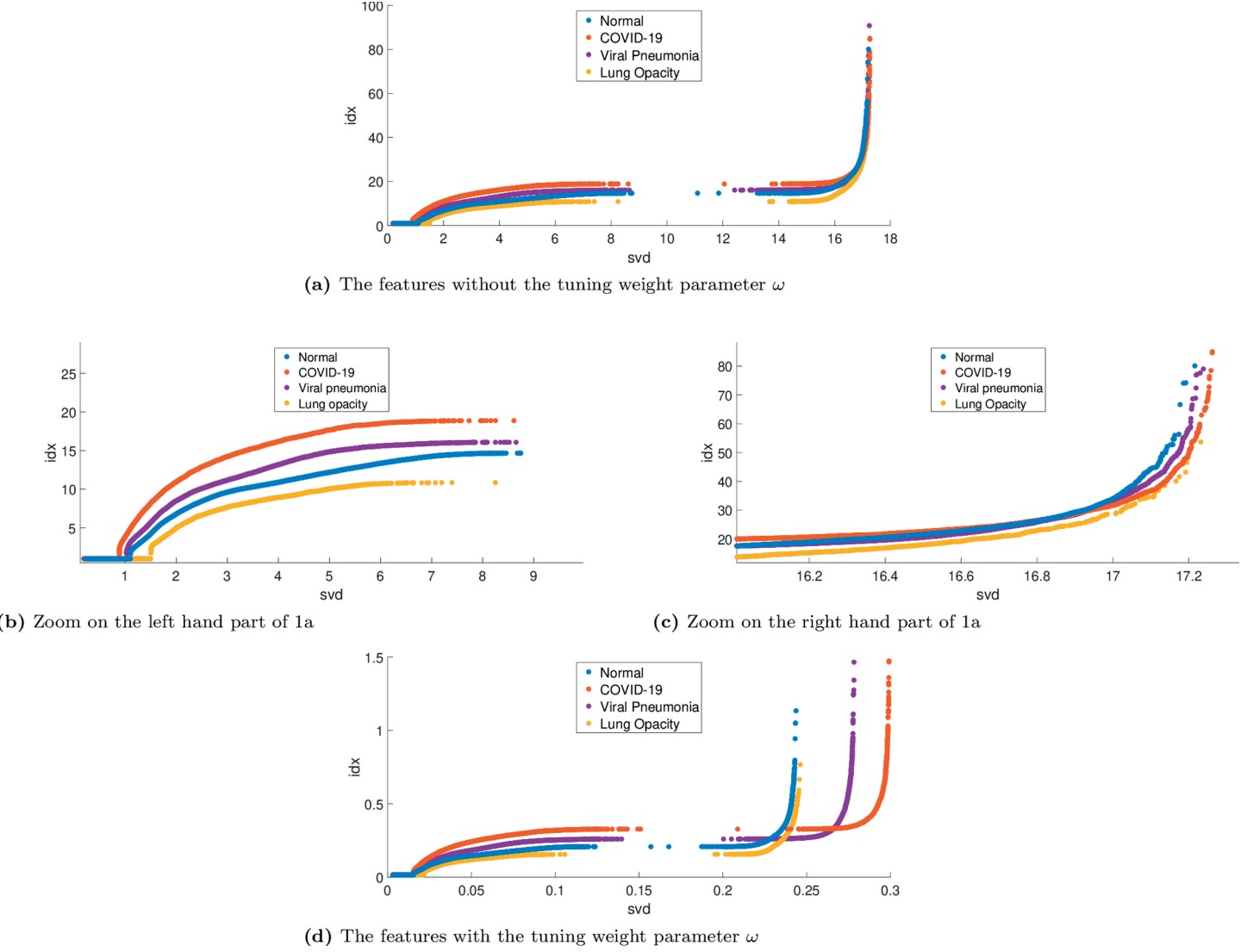

**Fig 6. Scatter plot of the feature vector using the complete dataset: Class 0 (normal, blue dots), Class 1 (COVID-19, red dots), Class 2 (viral pneumonia, purple dots), and Class 3 (lung opacity, yellow dots).** (a) Without the tuning weight parameter $\omega$. (d) With the tuning weight parameter $\omega$. The plots have similar shapes but exhibit a drift that aids in differentiating classes.

The rescaled feature vector $\rho_2$ (i.e., multiplied by the tuning weight) improved the distinction between the four classes, especially COVID-19, as shown in Fig 6(d). The tuning weight $\omega$ was found equal to $\{0.0142; 0.0173; 0.0143; 0.0161\}$, respectively for the normal, COVID-19, viral pneumonia, and lung opacity classes. The singular values $\zeta_k$ and the conditional indices $\eta_k$ were found in the ranges $[0.003, 0.267]$ and $[0.061, 1.208]$, respectively. The rescaling mapped the features onto a different scale, where the classes are better discriminated. This rescaling significantly improved the performance of the ensemble learning classifier. S4 Fig and S5 Fig Figs, in the "Supporting Information" section, show the confusion matrices for training and testing, respectively. These results are aligned with the observed class separation shown in Fig 6(d).

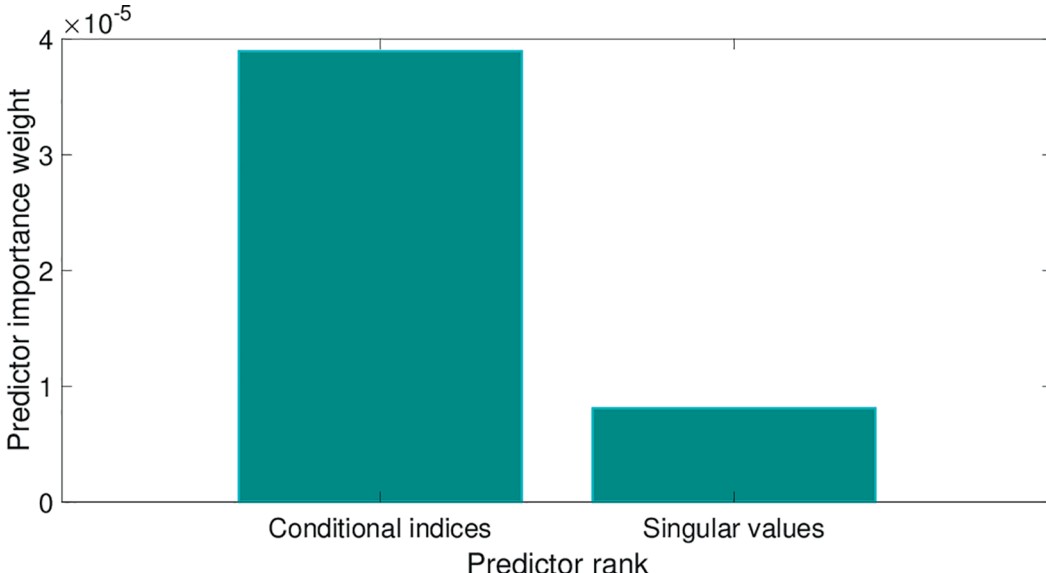

**(a)** Without tuning weight parameter $\omega$

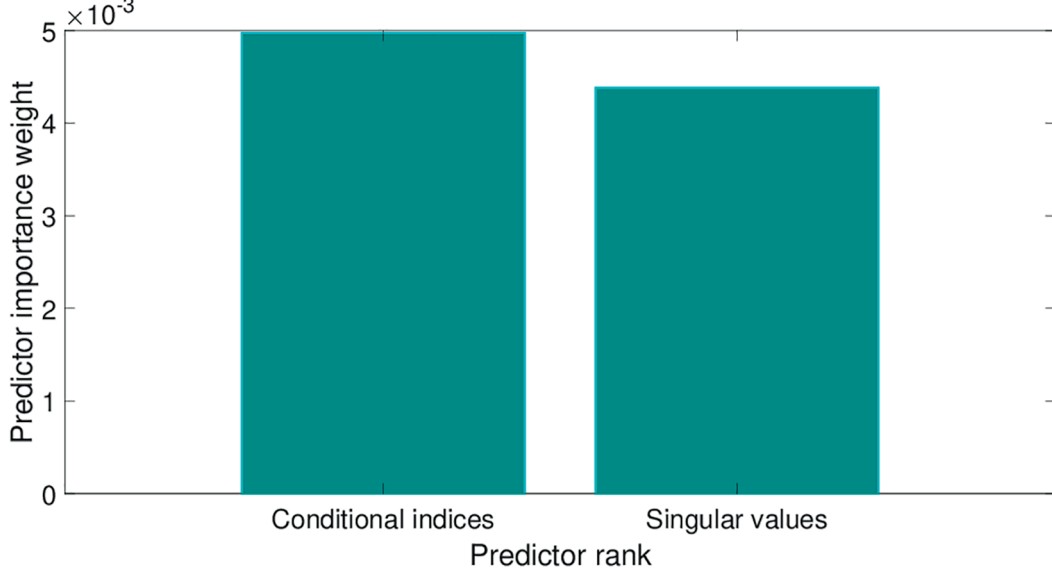

**(b)** With tuning weight parameter $\omega$

**Fig 7. Relative feature importance computed by ReliefF using the complete dataset.** (**a**) Without the tuning weight parameter $\omega$. (**b**) With the tuning weight parameter $\omega$.

Figs 8 and 9 show the partial dependence plots of the two features, without and with the tuning weight. The curves show the effect of the features on predicting each class separately. Figs 8(a) and 8(b) show the compared effect of the tuning weight with the conditional indices on the model's behaviour. We can observe that the model has similar behaviour for conditional indices between 0 and 20 and their corresponding weighted values. In this range, the model favours the class "lung opacity" (till index 8), then has more confidence in predicting

**Table 3. Thresholds for all data with and without the tuning weight parameter ($\omega$). Feature values for SVD ($\zeta$) and the conditional index ($\eta$) are different and can serve with a threshold detection scheme.**

| Feature | Chest X-ray image | | | | |
|---|---|---|---|---|---|
| | | Normal | COVID-19 | Viral pneumonia | Lung Opacity |
| $\zeta$ | Mean | 5.084 | 4.769 | 5.138 | 4.900 |
| | std | 5.748 | 6.031 | 5.616 | 5.895 |
| | Bounds | [0.213, 17.215] | [0.203, 17.261] | [0.320, 17.231] | [0.189, 17.243] |
| $\eta$ | Mean | 8.926 | 11.557 | 7.386 | 9.978 |
| | std | 7.472 | 9.852 | 5.808 | 8.469 |
| | Bounds | [1, 80.097] | [1, 85.025] | [1, 53.669] | [1, 90.817] |
| $\omega\zeta$ | Mean | 0.071 | 0.082 | 0.073 | 0.079 |
| | std | 0.081 | 0.104 | 0.080 | 0.095 |
| | Bounds | [0.003, 0.243] | [0.003, 0.299] | [0.004, 0.246] | [0.003, 0.278] |
| $\omega\eta$ | Mean | 0.126 | 0.200 | 0.105 | 0.160 |
| | std | 0.105 | 0.170 | 0.082 | 0.136 |
| | Bounds | [0.014, 1.133] | [0.017, 1.47] | [0.014, 0.766] | [0.016, 1.465] |

**Table 4. Performace metrics average: TNR- True Positive Rate (or recall, or sensitivity), FNR- False Negative Rate (FNR), PPV- Positive Predictive Values, FDR- False Discovery Rate, AUC- Area Under Curve, ACC- Accuracy rate, and Total Cost.**

| | TPR | FNR | ACC | AUC | PPV | FDR | Cost |
|---|---|---|---|---|---|---|---|
| **Learning without tuning** | 84.5 | 15.4 | 88.9 | 0.97 | 84.5 | 15.4 | 10573 |
| **Testing without tuning** | 84.4 | 15.5 | 88.7 | 0.97 | 84.3 | 15.6 | 1191 |
| **Learning with tuning** | 99.9 | 0.1 | 99.9 | 0.99 | 99.9 | 0.1 | 37 |
| **Testing with tuning** | 99.9 | 0.1 | 99.9 | 0.99 | 99.9 | 0.1 | 1 |

"viral pneumonia" (between 8 and 18), and then switches to predicting "COVID-19" with high confidence (index around 18). However, after applying the tuning weight, the effect is obvious: the model predicts "COVID-19" with high confidence and high probability, starting from the weighted value of 0.02. This observation aligns with our results, showing that the classification model with weighted features performs significantly better.

Similarly, when comparing Figs 9(a) and 9(b), the model behaves similarly with and without the tuning weight for singular values between 0 and 4. The model first favours "COVID-19" in this range with significant confidence. Then, it becomes more confident in predicting the "normal chest" class with significant probability. However, with the weighted singular values, the model predicts the "normal chest" class with high confidence and probability.

Using the jointly weighted conditional indices and the weighted singular values, it appears clear that the model finds a combined pattern that allows it to differentiate the four classes and, more importantly, "COVID-19" from "normal chest".

Considering all the plots from Figs 8 and 9, it is reasonable to hypothesise that using the four features together will be useful for learning complex image patterns. However, we did not test this hypothesis in this study, as the results were good with only the weighted features.

In texture analysis, the classification approaches are based on spatial localisation used in methods of edge detection and discrimination function features. Spatial localisation's main challenge lies in the fact that it is difficult to distinguish the boundaries of the texture and the micro-edge found in the same texture. At the same time, the discrimination functions depend on the discriminative capacity of its texture characteristics. This way, the X-ray image class types can be defined based on anatomical or disease levels and include several cell and tissue types. This work specifically focuses on the characteristics given by singular values and conditional indices, which allows us to suggest that the features under study can be used for texture

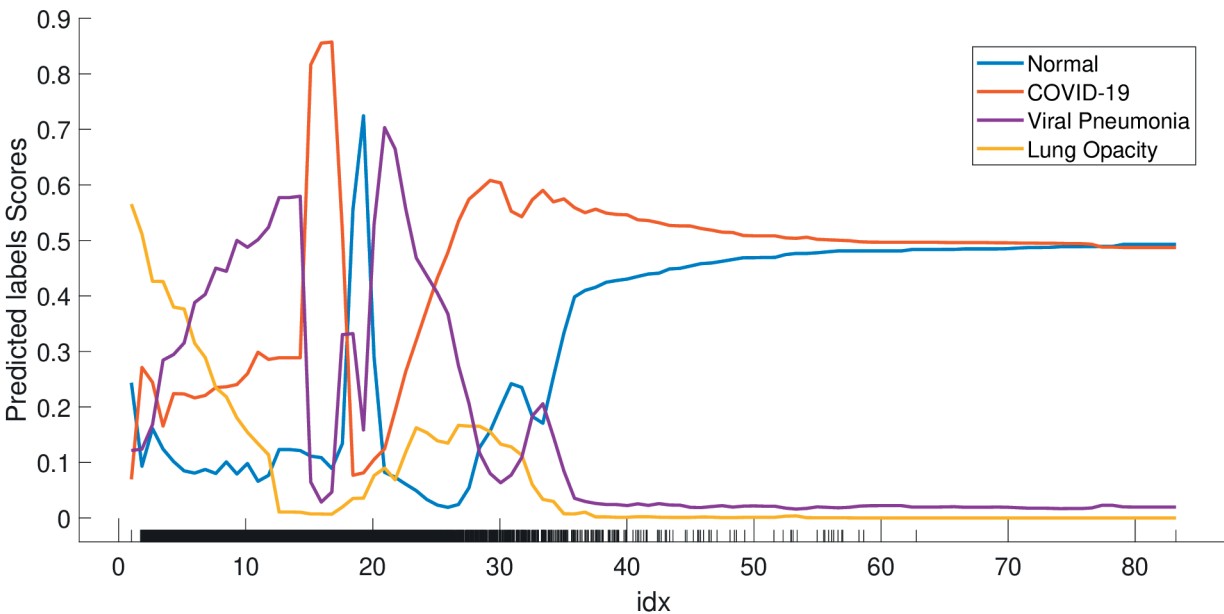

**(a)** Conditional indices without $\omega$

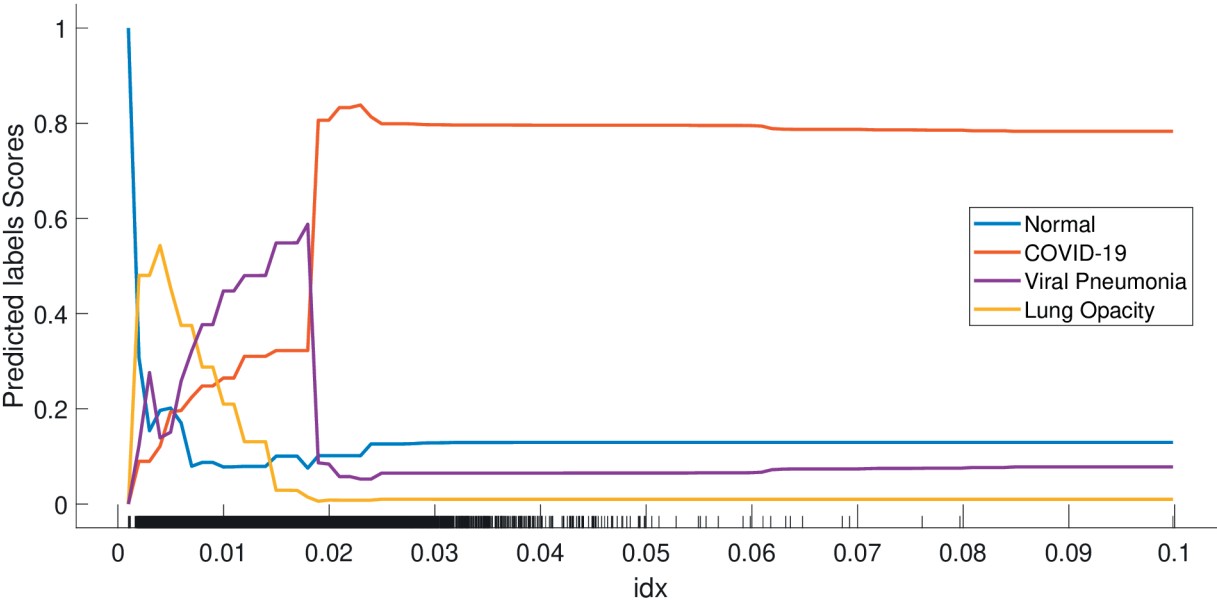

**(b)** Conditional indices with $\omega$

**Fig 8. Partial dependency plots of conditional indices during testing, with and without the tuning weight $\omega$: Class 0 (normal, blue dots), Class 1 (COVID-19, red dots), Class 2 (viral pneumonia, purple dots), and Class 3 (lung opacity, yellow dots).** The x-axis values are the conditional indices in (a) and the weighted conditional indices in (b). Values of the y-axis are probabilities of predicting a class. For example, in (a), the probability of predicting COVID-19 is 0.9 when the conditional index is around 18. Curves show the variation of the probabilities of predicting each class depending on the values of the feature. Figure (b) shows that for values of the weighted conditional indices starting from 0.2, the model is highly confident in predicting COVID-19, with a high probability (more than 0.8).

discrimination between different classes. Table 5 shows some examples of classifiers and feature extraction for X-ray images. Remarkably, our method scores are higher than CNN-based deep learning methods, which are currently at the forefront of X-ray image classification. They

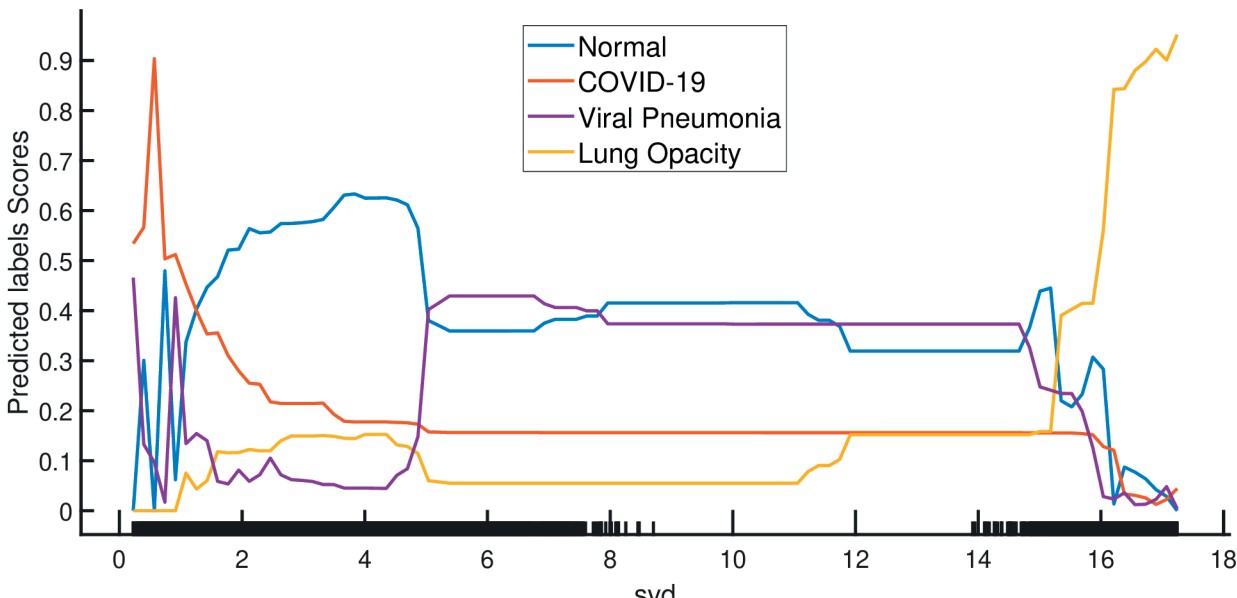

**(a)** Singular values without $\omega$

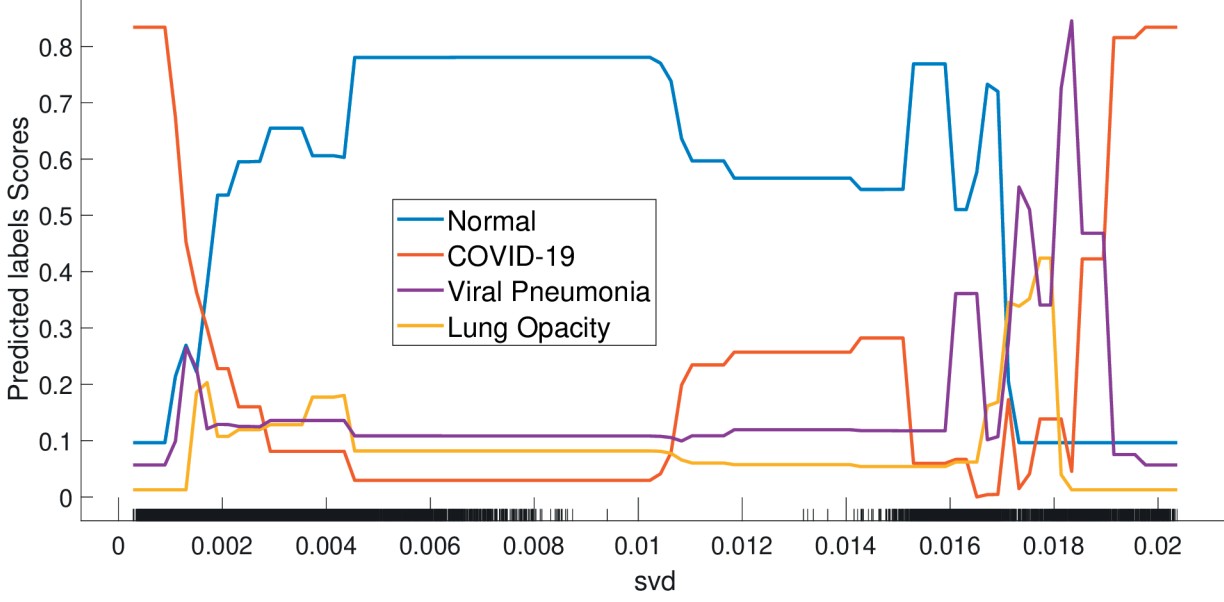

**(b)** Singular values with $\omega$

**Fig 9. Partial dependency plots of singular values during testing, with and without the tuning weight $\omega$: Class 0 (normal, blue dots), class 1 (COVID-19, red dots), class 2 (viral pneumonia, purple dots), and class 3 (lung opacity, yellow dots).** The x-axis values are singular values in (a) and weighted singular values in (b). Values of the y-axis are probabilities of predicting a class. For example, in (a), the probability of predicting COVID-19 is 0.9 when the singular value is around 0.40. Curves show the variation of the probabilities of predicting each class depending on the values of the feature. Comparing plots in (a) and (b), we notice that the behaviour of the model is similar in the range $[0, 5]$. However, Figure (b) shows that for the weighted singular values between $[0.005, 0.010]$, the model is highly confident in predicting normal chest, with high probability (close to 0.8).

generally outperform conventional feature-based methods but with higher computational costs. The reader is referred to [51–53] for a recent review about deep learning models for chest disease detection using X-ray images.

**Table 5. State-of-the-art methods for X-ray image classification. Summarised in terms of the classifier, preprocessing, and features extraction used and their performance using the different datasets. CLAHE: Contrast limited adaptive histogram equalization. DT: Decision Tree, HOG: Histogram of Oriented Gradients, WMF: Weighted Median Filtering, LSTM: Long short-term memory. PWLGBP: Weighted Local Gabor Binary Pattern. ENNSA: Ensemble Neural Net Sentinel Algorithm. IGLCM: Insistent Grey Level Co-occurrence Matrix. DF-GAN: Deep Fusion Generative Adversarial Networks. The performance metrics are the True Positive Rate, recall or Sensitivity (TPR), the True Negative Rate, Negative Recall, or Specificity (TNR), and the Accuracy Rate (ACC).**

| Classifier | Preprocessing | Features | TPR | TNR | ACC | Dataset | Ref. |
|---|---|---|---|---|---|---|---|
| Ensemble Bagged | Grayscale image transformation | singular values + conditional indices + variance-decomposition proportion | 0.99 | 0.99 | 0.99 | [28] | our |
| Ensemble Bagged | Grayscale image transformation | singular values + conditional indices | 0.84 | 0.75 | 0.88 | [28] | our |
| CNN | Image resizing and Augmentation | DenseNet | 0.99 | 0.99 | 0.99 | [28] | [29] |
| CNN | Image resizing | Custom | 0.97 | 0.98 | 0.96 | [28] | [54] |
| SVM | - | Mobilnetv2 | - | - | 0.98 | [28] | [55] |
| Quadratic SVM | - | DenseNet | 0.96 | 0.96 | 0.96 | [28] | [30] |
| CNN | Image resizing | U-Net | 0.94 | 0.95 | 0.95 | [28] | [30] |
| Decision Trees + PCA + Simple ANN | Adaptive Contrast Enhancement + Adaptive Local Thresholding | SqueezeNet | 0.94 | 0.97 | 0.92 | [28] | [56] |
| SVM, kNN, DT | | HOG | 0.97 | 0.98 | 0.98 | | [57] |
| Adaptive Attention Network | - | ResNet | 0.94 | 0.96 | 0.94 | [58,59] | [60] |
| CNN | Sobel+ filter | CovMnet | 1.00 | 0.98 | 0.97 | [28] | [61] |
| DCCNet | shifts, flips, zooms, crops and rotations | CNN + HOG | 0.97 | - | 0.98 | [62] | [63] |
| CNN | Gaussian smoothing + CLAHE | VGG-16 | 0.95 | - | 0.95 | [58] | [64] |
| CNN Fusion | Scaling, normalization, and augmentation | ResNet50V2, VGG16, InceptionV3 | 0.98 | 0.99 | 0.96 | [62] | [65] |
| SVM | Resized | CNN | 0.97 | 0.99 | 0.99 | [28] | [66] |
| Ensemble Bagged | WMF + segmentation + PWLGBP | LSTM | - | - | 0.87 | [62] | [67] |
| ENNSA | Median filtering + segmentation | IGLCM | 0.49 | - | 0.99 | [62] | [68] |
| Vision-transformer | Patches | EfficientNet | 0.98 | 0.99 | 0.99 | [62,69,70] | [12] |
| VGG16 + ResNet50 | DF-GAN data augmentation | Mask map | 0.86 | 0.86 | 0.87 | [71,72] | [73] |

# 4 Conclusions

This work proposed an original method to classify chest X-ray images corresponding to different diseases, such as COVID-19, viral pneumonia, lung opacity, and normal. The proposed method is based on two features estimated using an SVD decomposition. These are the singular values and the condition indices. Traditionally, these parameters are used for multi-collinearity diagnosis in case of regression. This work uses them as features to characterise the image texture. In addition, we introduced a tuning weight parameter to consider the variability of the attenuation of X-ray tissues. This weight is estimated using the coefficient of variation of the minimum covariance determinant from the bandwidth of the non-parametric distribution of variance-decomposition proportions. The resulting features were used with an ensemble learning method to classify normal, COVID-19, viral pneumonia, and lung opacity X-ray images. Performance was assessed using True Positive Rate, False Negative Rate, Positive Predictive Values, False Discovery Rate, Area Under Curve, Accuracy Rate, and Total Cost. On average, the method achieved an accuracy of 88% without applying the tuning weight and 99% with it. This result suggests that the proposed method can be used as an

efficient texture discriminator to characterise different tissues in X-ray images, especially in respiratory syndromes.

In addition to its excellent performance, the proposed method has a descent computational cost with imbalanced data, compared to current deep learning methods. Computer-based quantitative image texture analysis has an important potential to improve image interpretation by yielding reproducible results.

The proposed method's main limitations are that it does not explicitly consider physiological and non-physiological artefacts, clutter, and ambiguities in the images, and it is difficult to deal with highly imbalanced data. In the medical context, there is a lack of clear definitions of biomedical texture information for validation and translation to routine clinical applications. There is also a lack of an appropriate framework for multiscale and multispectral analysis in 2D and 3D images. Computer-based quantitative imaging features can be challenging to interpret, as they can appear abstract to a physician, and their meaning in the clinical context may not be directly apparent.

Future work will focus on a more extensive evaluation of the proposed approach, the study of robust feature extraction methods using an approximation of the image by a sparse low-rank matrix [74–77]. Additional work can target image segmentation coupled with soft tissue decomposition to identify tissue-at-risk regions affected by COVID-19 and other respiratory pathologies, where all lung images exhibit patches. Also, it allows studying the taste receptors and their relationship with the walls of blood vessels in the lungs and the characterization of acute respiratory distress syndrome (ARDS) [78].

## Supporting information

**S1 Fig**. **Confusion Matrices without the tuning weight parameter during training for Class 0 (normal), Class 1 (COVID-19), Class 2 (viral pneumonia), and Class 3 (lung opacity).** (PDF)

**S2 Fig**. **Confusion Matrices without the tuning weight during testing for Class 0 (normal), Class 1 (COVID-19), Class 2 (viral pneumonia), and Class 3 (lung opacity).** (PDF)

**S3 Fig**. **Receiver Operating Characteristic (ROC) and Area Under the Curve (AUC) of the features without the tuning weight $\omega$.** (PDF)

**S4 Fig**. **Confusion Matrices with the tuning weight parameter during training for Class 0 (normal), Class 1 (COVID-19), Class 2 (viral pneumonia), and Class 3 (lung opacity).** (PDF)

**S5 Fig**. **Confusion Matrices with the tuning weight parameter during testing for Class 0 (normal), Class 1 (COVID-19), Class 2 (viral pneumonia), and Class 3 (lung opacity).** (PDF)

To reproduce these results, please use:https://github.com/tonioquinterorincon/Computer-based-quantitative-image-texture-analysis-using-multi-collinearity-diagnosis-in-chest-X-ray

## Acknowledgments

The authors are grateful to the Faculty of Engineering and Agricultural Sciences at the Pontifical Catholic University of Argentina (UCA) for providing financial support for the publication of this work.

## Author contributions

**Conceptualization:** Antonio Quintero-Rincon, Ricardo Di-Pasquale, Karina Quintero-Rodríguez, Hadj Batatia.

**Data curation:** Antonio Quintero-Rincon.

**Formal analysis:** Antonio Quintero-Rincon, Hadj Batatia.

**Investigation:** Antonio Quintero-Rincon.

**Methodology:** Antonio Quintero-Rincon, Ricardo Di-Pasquale, Karina Quintero-Rodríguez, Hadj Batatia.

**Project administration:** Antonio Quintero-Rincon.

**Resources:** Antonio Quintero-Rincon, Ricardo Di-Pasquale.

**Software:** Antonio Quintero-Rincon.

**Validation:** Antonio Quintero-Rincon, Karina Quintero-Rodríguez, Hadj Batatia.

**Visualization:** Antonio Quintero-Rincon, Hadj Batatia.

**Writing – original draft:** Antonio Quintero-Rincon, Hadj Batatia.

**Writing – review & editing:** Antonio Quintero-Rincon, Ricardo Di-Pasquale, Karina Quintero-Rodríguez, Hadj Batatia.

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
