## [Decision Letter · Decision Letter 0]

16 Jan 2025

PONE-D-24-57634Computer-based quantitative image texture analysis using multi-collinearity diagnosis in chest X-ray imagesPLOS ONE

Dear Dr. Quintero-Rincon,

Thank you for submitting your manuscript to PLOS ONE. After careful consideration, we feel that it has merit but does not fully meet PLOS ONE’s publication criteria as it currently stands. Therefore, we invite you to submit a revised version of the manuscript that addresses the points raised during the review process.

We look forward to receiving your revised manuscript.

Kind regards,

Khan Bahadar Khan, Ph.D

Academic Editor

PLOS ONE

Journal Requirements:

2. Please note that in order to use the direct billing option the corresponding author must be affiliated with the chosen institute. Please either amend your manuscript to change the affiliation or corresponding author, or email us at plosone@plos.org with a request to remove this option.

Reviewers' comments:

Reviewer's Responses to Questions

**Comments to the Author**

1. Is the manuscript technically sound, and do the data support the conclusions?

Reviewer #1: Yes

Reviewer #2: Yes

Reviewer #3: Yes

2. Has the statistical analysis been performed appropriately and rigorously? 

Reviewer #1: Yes

Reviewer #2: Yes

Reviewer #3: Yes

3. Have the authors made all data underlying the findings in their manuscript fully available?

Reviewer #1: Yes

Reviewer #2: Yes

Reviewer #3: Yes

4. Is the manuscript presented in an intelligible fashion and written in standard English?

Reviewer #1: Yes

Reviewer #2: Yes

Reviewer #3: Yes

5. Review Comments to the Author

Reviewer #1: Authors have purposed a method to classify viral infection based chest X-ray images based on the singular values and the condition indices to characterize image texture using an SVD decomposition. It is noteworthy that the proposed method has low computational cost with imbalanced data as compared to other methods and good performance as measured by TPR, FNR, FDR, PPV, AUC, etc. With the emerging use of computer based quantitative image texture analysis for comprehensive and reproducible image analysis, this manuscripts adds value to the research community. I recommend to publish this manuscript in the current status.

Reviewer #2: The authors introduces a novel method for automatic classification of chest X-ray images, aiming to differentiate between four categories: normal, COVID-19, viral pneumonia, and lung opacity. The paper proposes the use of singular values and conditional indices, extracted through SVD, as unique features to characterize tissue texture in chest X-rays. Traditionally, these parameters are used to diagnose multi-collinearity in regression models, but here they are adapted for texture analysis. To account for variability in X-ray attenuation caused by pathological changes, a tuning weight parameter, ω, is introduced. This parameter is derived using the coefficient of variation from the minimum covariance determinant of variance-decomposition proportions. The extracted features, weighted by ω are used in an ensemble bagged trees classification model. This approach addresses class imbalance effectively. In the results without weight tuning, conditional indices are shown to be the more significant predictors for classification. When weight tuning is included, singular values gain importance, and the overall separation between classes improves, particularly for COVID-19. The proposed method demonstrates high accuracy and computational efficiency, making it a promising tool for automated diagnosis of respiratory conditions in chest X-rays, specifically between COVID-19 and viral pneumonia, which can be hard to distinguish in chest X-rays.

Although I was able to follow the manuscript, it might need the help of a proofreading service to meet publication standards. Some of them are enumerated below, but my list is not exhaustive.

Multiple instances of figure and figure references out of order throughout paper. Figures should be numbered consecutively in the order they are mentioned in the text. Move references to figures to match order of figure placement in body of paper, or move placement and order of figures to match reference order.

Are Figures 10-14 part of the S1 appendix or part of main body? Supplementary figures are referred to as "Supporting Information" and should be numbered with an "S" prefix followed by a numeral (e.g., S1 Fig, S2 Fig). If Figure 10-14 are Supporting Information, update the caption and figure numbering appropriately and this would take care of several out of order figure references.

Figure 4 is referenced (Line 130) before Figure 1-2 (Line 148).

Figure 4 is referenced in body (Line 182) before Figure 3 (Line 213), but Figure 3 is placed before Figure 4 in body.

References to Figure 10 and Figure 11 (lines 369-370) occur before references to Figure 8 and Figure 9 (lines 383-384). References to Figure 13 and Figure 14 (lines 380) occur before references to Figure 8 and Figure 9 (lines 383-384). If these are SI figures, update caption and figure numbering and figure order is correct.

Figure 12 is not referenced in paper at all, nor discussed in results.

Having the S1 Appendix figure occur in the middle of the references makes it difficult to read the references and papers.

Figure 5 color palette and symbol/line choice does not make the data easy to read and interpret. I would suggest to update the color palette for the data set to match Figures 6, 8, 9 for consistency. In addition, use different symbols/lines for data vs fit to make it easier to read the data. The color palette and use of solid lines for every data set made it hard to interpret the results.

The density curves in Figure 5 do not show any statistical variances in the fit curves for Normal, COVID 19, and LungOp. Is this to be expected? Is this indicative of anything? How does the lack of variance in the curves affect the tuning weight parameter ω?

The wording of Line 378-379 does not make sense, what is the author trying to say.

Figure 8 and Figure 9 are near identical and when overlaid on top of each other do not show any meaningful differences. Table 2 shows statistical differences between the singular and conditional values. I would therefore expect a difference in the predicted label scores for each parameter. Explain why the plots for the predicted table scores for the singular and conditional data should be near exactly the same or update plots to show how the data is different.

I would suggest updating order of datasets and legends to be consistent through out all figures. The order of the data in the figures change throughout the paper depending on the figure. Suggest using a consistent order in all figures, i.e. Normal, COVID-19, Viral Pneumonia, and Lung Opacity, so that reader doesn’t have to read caption to determine order of data in each figure.

In Figure 12, spell out the acronyms for ROC and AUC

Reviewer #3: This paper introduces a novel algorithm that leverages features derived from singular value decomposition (SVD) for image texture analysis, specifically targeting the detection of abnormal X-ray images based on tissue attenuation. While the idea of using SVD for feature extraction has been explored in the past, the authors address the critical challenge of constructing meaningful and effective features from the decomposition. They propose using singular values and conditional indices as texture features, combined with a newly introduced tuning-weight parameter. This parameter, estimated through the coefficient of variation of the minimum covariance determinant from the variance-decomposition proportions of the SVD, accounts for the variability in tissue attenuation affected by pathologies.

The paper validates the proposed methodology using a challenging chest X-ray dataset with imbalanced classes, including COVID-19, viral pneumonia, lung opacity, and normal cases. The use of an ensemble bagged trees multiclass classification method achieved impressive accuracy rates: 88% without the tuning weight and 99% with it. The authors convincingly show that the proposed features, when enhanced by the tuning weight, improve key performance metrics such as True Positive Rate, False Negative Rate, Positive Predictive Value, False Discovery Rate, Area Under the Curve, Accuracy Rate, and Total Cost. Figures 6 and 7 clearly highlight the robustness of their approach.

The algorithms are well-documented, and the experimental results are thoroughly analyzed. The results are interesting and the paper is well-written. Only minor points require the authors' attention, if feasible.

• For readers who are not familiar with singular value decomposition (SVD) and principal component analysis (PCA), the authors should provide a brief explanation of how PCA approximates the original features. Specifically, it would be helpful to clarify the relationship between the original features and their lower-dimensional representation using PCA.

• The paper mentions that decision-making is based on identifying near dependencies between the columns of the image matrix X. This naturally raises the question of whether analyzing the near dependencies between the rows (by transposing X and applying the same analysis) might yield additional insights. Have the authors explored this approach? If so, what were the results?

• In Figure 4, the term "tuning width" is mentioned but not clearly explained. What does this process involve, and how does it contribute to the proposed methodology?

• The x-axis labels in Figures 8 and 9 are unclear. Since these figures play a crucial role in showing the robustness of the proposed approach, the authors should provide more detailed explanations. In addition to the statement, "This feature combination suggests that these predictors are a powerful tool for discriminating between different classes of chest X-ray images," please elaborate on the specific judgments or insights that can be drawn from these figures.

• In the conclusion, the paper claims, "In addition to its excellent performance, the proposed method has a low computational cost with imbalanced data, compared to other methods." However, this statement is difficult to agree with, given that performing SVD is computationally expensive, especially for large matrices. Although low-rank approximations of SVD can reduce costs, the paper does not clarify whether this was used. If the proposed method is indeed computationally efficient, please specify the baseline or methods to which it is being compared. Most of the conventional feature extraction methods listed in Table 1 are not computationally expensive, so this comparison needs further clarification.

Typos

• quatify: in the first sentence of section 2.2 Methodology

• there are several places where ‘comma’ is missing before ‘where’

• inconsistency in placing a space before ‘where’

6. PLOS authors have the option to publish the peer review history of their article (what does this mean?). If published, this will include your full peer review and any attached files.

Reviewer #1: No

Reviewer #2: No

Reviewer #3: No

---

## [Author Response · Author response to Decision Letter 1]

8 Feb 2025

We thank the editor and the three reviewers for their careful and thoughtful examination of our paper. We appreciate the time and effort they have put into their reviews. The corresponding PDFs have been attached: PONE-D-24-57634 Response to Reviewers, PONE-D-24-57634 Revised Manuscript with Track Changes, Manuscript, and latex fonts. The main changes made to the manuscript are shown in red. We also proofread the whole text and made many changes (not shown in red) to correct some typos and improve the writing style for better readability.

---

## [Decision Letter · Decision Letter 1]

23 Feb 2025

Computer-based quantitative image texture analysis using multi-collinearity diagnosis in chest X-ray images

PONE-D-24-57634R1

Dear Dr. Quintero-Rincon,

We’re pleased to inform you that your manuscript has been judged scientifically suitable for publication and will be formally accepted for publication once it meets all outstanding technical requirements.

Kind regards,

Khan Bahadar Khan, Ph.D

Academic Editor

PLOS ONE

Additional Editor Comments (optional):

Reviewers' comments:

Reviewer's Responses to Questions

**Comments to the Author**

1. If the authors have adequately addressed your comments raised in a previous round of review and you feel that this manuscript is now acceptable for publication, you may indicate that here to bypass the “Comments to the Author” section, enter your conflict of interest statement in the “Confidential to Editor” section, and submit your "Accept" recommendation.

Reviewer #1: All comments have been addressed

Reviewer #2: All comments have been addressed

Reviewer #3: All comments have been addressed

2. Is the manuscript technically sound, and do the data support the conclusions?

Reviewer #1: Yes

Reviewer #2: Yes

Reviewer #3: Yes

3. Has the statistical analysis been performed appropriately and rigorously? 

Reviewer #1: Yes

Reviewer #2: Yes

Reviewer #3: Yes

4. Have the authors made all data underlying the findings in their manuscript fully available?

Reviewer #1: Yes

Reviewer #2: Yes

Reviewer #3: Yes

5. Is the manuscript presented in an intelligible fashion and written in standard English?

Reviewer #1: Yes

Reviewer #2: Yes

Reviewer #3: Yes

6. Review Comments to the Author

Reviewer #1: Authors have clarified some of the comments. I have no additional questions. I recommend for its publication.

Reviewer #2: I appreciate the authors' effort in improving the manuscript and dedication to the peer-review process. Based on the red-inked version, it looks like the messaging is clearer and all my concerns were addressed.

Reviewer #3: The revised version of this manuscript takes into account my previous remarks and objections.

All the suggested requests are clearly explained and presented.

Authors added proper and sufficient explanations and added more numbers to clear things.

I believe that this manuscript is ready to be published.

7. PLOS authors have the option to publish the peer review history of their article (what does this mean?). If published, this will include your full peer review and any attached files.

Reviewer #1: No

Reviewer #2: No

Reviewer #3: **Yes: **Eunjung Lee

---

## [Editor Report · Acceptance letter]

PONE-D-24-57634R1

PLOS ONE

Dear Dr. Quintero-Rincon,

I'm pleased to inform you that your manuscript has been deemed suitable for publication in PLOS ONE. Congratulations! Your manuscript is now being handed over to our production team.

Kind regards,

on behalf of

Dr. Khan Bahadar Khan

Academic Editor

PLOS ONE